# Position: Generative AI and Differential Privacy — A Perfect Match

## Abstract

Generative Artificial Intelligence (GenAI) has evolved into a transformative technology whose unprecedented growth and public exposure have revealed challenging issues ranging from privacy protection to reducing factual inaccuracies and hallucinations, model security risks, legal complications, and a lack of interpretability. This *position paper* examines how Differential Privacy (DP), a mathematical privacy protection framework, can address both privacy concerns *and* other systemic challenges beyond privacy in GenAI. We argue that DP is a versatile and underutilized tool with significant potential to address many critical GenAI issues. To argue our claim, we connect the core principle of DP to these issues, evaluate existing research, and pose relevant research questions.

## 1. Introduction

GenAI has witnessed unprecedented growth in recent years, evolving from a research endeavor to generating real-world value. From producing realistic images to crafting human-like text, GenAI continues to push technological boundaries. However, its rapid adoption has exposed major challenges, as issues that were once mainly of interest to the research community have now emerged as serious problems. Some issues relate to the development of more powerful GenAI systems, while others focus on ethical considerations vital for both humanitarian reasons and maintaining public trust. Additionally, concerns around reliability and security are crucial for advancing GenAI into safety-critical domains.

Differential Privacy (DP) is a mathematical framework that addresses privacy protection, offering strong theoretical guarantees (Dwork & Roth, 2014). At its core, (user-level) DP ensures that the inclusion or exclusion of *any single* user's data has an (almost) negligible impact on the outcome of a computation. This property makes it nearly impossible to infer whether a specific individual contributed to the computation, thereby safeguarding their privacy. In this position paper, we argue that **DP is a (1) highly versatile tool with (2) significant untapped potential to resolve current GenAI issues**.

DP may initially seem unrelated to challenges beyond the realm of privacy; however, its versatility, which stems from its minimal application requirements, allows it to be deployed across various domains and problems (Desfontaines, 2021). Nonetheless, this only implies that DP can *technically* be used to solve many issues. What *justifies* its use is that its underlying paradigm—ensuring that no single input disproportionately influences the output—is a desirable quality across a wide range of applications and problems. Additionally, unlike many other approaches, DP offers mathematically rigorous and reliable guarantees, making it the ideal complement to other, more heuristic methods.

Drawing inspiration from (Hendrycks et al., 2021; Nah et al., 2023; Singla et al., 2024), we identified key issues, concentrating on areas where the application of DP is most relevant and impactful. Concretely, the issues are backdoor attacks (Appendix A due to space constraints), data quality challenges, harmful content generation, memorization of Personally Identifiable Information (PII), hallucinations, the lack of interpretability, exhaustion of high-quality data, and the "right to erasure".

We argue our position's *first* claim by demonstrating how DP can be applied to address a wide spectrum of challenges in GenAI. Where prior research exists, we draw on empirical findings to support our analysis. Yet, we find many applications of DP in GenAI remain underexplored, with little to no existing literature addressing these intersections; this is direct evidence that supports our *second* claim. To address this gap, we pose important unanswered questions and propose future research directions. **The goal of our paper is to underline DP's untapped potential and inspire deeper exploration into its applications within GenAI**.

We group the memorization of PII, data quality, and harmful knowledge under the "Proactive Measures" cluster, based on their resolution methods, all of which apply before model training. Other issues are distinct enough to warrant a separate discussion.

The remainder of the paper is structured as follows. We begin with an informal background section on GenAI and DP. We then address the identified issues sequentially, progressing from those most closely aligned with DP's original privacy-focused objective to more speculative applications. Next, we critically examine our own position through the

lens of an opposing viewpoint. Finally, we synthesize these contrasting positions into a comprehensive conclusion.

## 2. Background

**Generative Artificial Intelligence.** GenAI employs Artificial Intelligence (AI) systems to develop generative models that learn and sample from high-dimensional probability distributions, enabling the creation of new data with statistical properties similar to the training set. These models either directly compute probability distributions or generate samples without explicit calculations. Examples include autoregressive Transformer models (Radford et al., 2018), which excel at generating coherent text sequences, and methods such as Generative Adversarial Networks (Goodfellow et al., 2014) and diffusion models (Croitoru et al., 2023), which iteratively transform noise into photorealistic image generation and artistic creation. These models can also be conditioned on input parameters, guiding applications ranging from interactive chatbots to controlled image generation.

**Differential Privacy.** Formally, a DP mechanism ensures that the outputs computed on *any* two datasets differing by only one data unit are very "close", making it (nearly) impossible to infer which dataset was used. The notion of "closeness" varies slightly depending on the specific definition of DP, with each variation defined by parameter(s) that control the degree of similarity and determine the strength of privacy guarantees (Mironov, 2017; Dong et al., 2019). To provide its privacy guarantees, DP typically adds noise to an output to obscure the presence of individual samples. The appropriate level of noise is essential; too much randomness degrades results, while too little weakens privacy. This constitutes the privacy-utility tradeoff. Additionally, for any given problem, there may be multiple ways to implement DP at the same privacy level, each leading to different degrees of utility degradation. For example, to compute a dataset's mean, one could use Local Differential Privacy (LDP) to make data points less distinct from one another *before* averaging or compute the mean first and then apply DP. The latter approach, which depends on a trustworthy central aggregator with access to raw data, provides enhanced utility but requires trust in the aggregator. In contrast, the former incurs greater utility loss but eliminates the need for a central aggregator. DP has two essential properties for privacy-preserving data analysis: The **post-processing immunity** ensures that once anything has been made differentially private, this privacy guarantee cannot be removed or weakened through subsequent operations, enabling unrestricted downstream analysis. **Group privacy** extends DP's definition of *single-entry* differences between datasets to datasets differing in *multiple* samples. Group privacy describes how DP's guarantees decay gracefully with the number of differing samples.

Differentially Private Stochastic Gradient Descent (DP-SGD) (Abadi et al., 2016) has emerged as the predominant method to train Machine Learning (ML) models under DP. In DP-SGD, gradients are clipped during backpropagation to bound the influence of a single training sample, followed by the addition of noise.

## 3. Proactive Measures

Training GenAI models requires vast amounts of data, mainly collected from the Internet by crawling websites for text and images, followed by a curation process. The data curation process not only reduces the size of the datasets, helping to minimize computational and storage requirements along with their associated costs, but it also serves two additional important purposes: improving data quality and managing potential risks related to harmful content.

**Data Quality.** The need for high-quality data is substantiated by research demonstrating the detrimental effects of label noise (Frenay & Verleysen, 2014) and feature inaccuracy (Budach et al., 2022). For generative models, higher quality training data translates into a more accurate representation, as measured by their benchmark performances.

**Harmful Content.** Harmful content encompasses information that, when disseminated or utilized, can lead to harm, either directly or indirectly. Both vision-based and text-based models may reproduce harmful content. Vision-based models are concerned with outputting depictions of racist, sexist, and violent concepts (Bird et al., 2023) while text-based models contend with the execution of illegal activities, the encouragement of unethical or unsafe actions, the promotion of harmful ideologies, and the spread of misinformation (Wang et al., 2023b).

**Memorization of PII.** Large Language Models (LLMs) have been shown to memorize individual samples from their training data containing sensitive information such as names, phone numbers, and email addresses belonging to real people (Carlini et al., 2021; 2023a). Similarly, image generation models produce images of real people. The unsolicited release of private information represents a violation of a person's right to privacy and may expose the model developer to litigation. Privacy concerns about the collection of user interaction data have led many companies (Mok, 2023) and even Italy (McCallum, 2023) to (temporarily) ban specific GenAI. Memorization can cause a model to reproduce verbatim copies (text) or near-duplicates (image) of the original training data, leading users to *inadvertently* stumble upon this data while interacting with the model as intended. It also enables *attackers* to reconstruct the training data by exploiting information encoded in the model's weights (Song & Namiot, 2023). State-of-the-Art (SotA) GenAI models, due to their size, are particularly vulnerable

to memorization (Carlini et al., 2023b), a concern that is expected to grow as models scale. Additionally, the identified memorization in ML models represents only a lower bound, suggesting that the actual extent could be much greater.

### 3.1. Heuristic filtering

Current approaches to resolving the three problems rely on heuristic filtering. To mitigate **memorization of PII**, duplicates in the training data are removed using techniques such as n-gram similarity detection (Lee et al., 2022), while PII is filtered using regular expressions, named entity recognition, or rule-based logic (Microsoft Corporation). Through OCR, these methods extend to images. **Harmful content** is often addressed using keyword-based identification, AI tools (AI, 2020), or using post-hoc methods (Section 4). Ensuring **data quality** depends on the data modality: for text, repetitive or inappropriate content is filtered (Dubey et al., 2024), whereas for images, classifiers and checksums help exclude undesirable content and samples with poor aesthetic quality (Meyer et al., 2024). Despite significant advancements over the years, heuristic filtering methods remain inherently imperfect. This is primarily because information can inherently be conveyed in countless ways. Furthermore, the vast size of modern datasets renders manual oversight impractical. Consequently, some low-quality or harmful samples inevitably "slip through the net", even under rigorous processes (Marsoof et al., 2023).

### 3.2. DP as a Complement to Heuristic Filtering

The core problem with heuristic filtering is that, as the saying goes, "One bad apple can spoil the barrel"; it may be sufficient for only a few "bad" samples to evade the filtering to significantly undermine model quality. This is because heuristic methods *do not guarantee* that removing undesired samples prevents a model from compensating by extracting more harmful information from the remaining harmful data. In contrast, DP provides just that guarantee; while DP does not completely eliminate the risk (and benefit) posed by those samples, it can at least limit it. Another advantage that DP provides is the ability to move beyond a binary choice between complete removal or retention of data points. Instead, it enables a more nuanced approach, allowing data to be "partially" removed based on customizable criteria, making it particularly advantageous for ensuring data quality, as it allows extracting potentially useful data even from samples of uncertain quality.

### 3.3. Prior Work

Previous work on the effectiveness of data poisoning (Goldblum et al., 2020) demonstrates that the inclusion of only a few samples is sufficient to significantly deteriorate a model's quality. Though these samples are adversarially chosen, they demonstrate the potential of a single sample.

### 3.4. Open Questions and Challenges

- **Performance vs. Harm Mitigation Tradeoff:** How do the improvements in handling harmful samples provided by DP compare to the decline in performance that DP causes? How does our proposed method of applying DP only on *suspected* harmful samples perform? What beneficial information is lost? Are innocent models, stripped of harmful content, still useful?
- **Memorization and Duplicate Data:** How does memorization in private models scale with the presence of duplicate data? DP constrains this scaling by imposing an upper bound through group privacy. However, the guarantees provided by DP are often overly conservative, as real-world scenarios tend to yield tighter bounds than predicted by DP.

## 4. The "Right to Erasure"

Regulations, such as the General Data Protection Regulation (GDPR) and similar laws in other countries, grant citizens the legal right, known as the "right to erasure", to have their data deleted from the storage of any data controller. The right poses a formidable challenge for ML models, as data controllers must not only delete an individual's data from their datasets but also ensure that all models trained on that individual's data are updated or revised accordingly. In ML training, adding a sample to a dataset influences the model weights–through model training–in complex and *untraceable* ways. Therefore, the only way to ensure that a trained model has completely forgotten a person's data is to *retrain the model from scratch* using the same dataset with that person's data excluded (Nguyen et al., 2022). Several approximations to the full retraining procedure, called "approximate unlearning", have been devised to avoid computational and monetary costs. However, these methods struggle with the stochasticity and incrementality of training, and the potential for catastrophic forgetting (Wang et al., 2024b)—a phenomenon in which a model forgets or unlearns more than the targeted data (French, 1999). These three problems often result in unlearned models achieving lower performance than fully retrained models.

**Current Approaches.** Two central approaches have emerged to remove individuals' data from already trained models: **Alignment** teaches models to *refuse* access to harmful content. It can also be used more broadly to better align model outputs with human preferences (Bai et al., 2022; Lee et al., 2023). However, it has proven fragile, with aligned models potentially being manipulated or "jailbroken" to reveal dangerous knowledge (Wei et al., 2023). **Unlearning** aims to *forget* knowledge of harmful content from the model, operating on the principle that a model unable to recall spe-

cific information cannot generate related responses (Cao & Yang, 2015). Yet, recent research (Łucki et al., 2024) has revealed significant limitations, including the challenge of completely eliminating specific information without risking catastrophic forgetting.

### 4.1. DP: Prevention over Remediation

Unlike alignment or unlearning methods, which act retroactively, DP works proactively by guaranteeing that any user's impact on a trained model is minimal. One could thus argue that if a user's data is undetectable, it is effectively the same as their data not being used. While it does not offer a complete erasure of individual data as defined by current laws, it represents a *practical* alternative—eliminating the need for expensive retraining and reducing the unreliability and collateral damage associated with post-hoc techniques—that we believe is worth considering. Furthermore, we hypothesize that it is easier to apply unlearning to a private model. Intuitively, a model that was constrained to learn less about individuals during training should, in principle, have less information to unlearn.

### 4.2. Prior Work

DP inherently satisfies the definition of unlearning proposed by Sekhari et al. (2021), providing "unlearning for free" when training with DP. Furthermore, Sekhari et al. (2021) establish bounds on the number of samples that can be unlearned before the model's performance degrades beyond a specified threshold. Building on this, Huang & Canonne (2023) tighten these bounds for any unlearning algorithm that does not rely on side information.

### 4.3. Open Questions and Challenges

- **Alignment:** How do DP and alignment interact? Is alignment effective in DP-trained models, and how does it compare to nonprivate models?
- **Unlearning:** Does our hypothesis hold up? Is it easier or more challenging to perform unlearning on private models? Which factors influence the efficiency and completeness of unlearning in private models?

## 5. Copyright Infringement

"Copyright infringement occurs when a copyrighted work is reproduced, distributed, ... or made into a derivative work without the permission of the copyright owner." (U.S. Copyright Office) Copyright infringement lawsuits against several developers of LLMs, image generation models, and coding assistants are currently brought on by visual artists (Andersen v. Stability AI Ltd., 2023), music publishers (Concord Music Group, Inc. v. Anthropic PBC, 2024), authors (Alter v. OpenAI Inc., 2023), and software developers

(Doe 1 v. GitHub, Inc., 2022) with plaintiffs claiming that their data was used by the model developers without permission or compensation.[1] In the U.S., model developers currently invoke the "fair use" exception in copyright law to justify using copyrighted material for training purposes. However, whether training ML models constitutes "fair use" remains an unsettled legal question (Quang, 2021; Dornis & Stober, 2024). Solutions to the dispute are urgently needed. If courts uphold copyrighted data use by model developers, artists may restrict access to their data. Conversely, if data holders win, model development could slow as developers become overly cautious in filtering copyrighted data, hindering innovation.

**The Plaintiff's Perspective**. In copyright infringement cases involving GenAI models, plaintiffs face the challenge of proving that their data has been misused. Misuse falls into two categories: either the data was used to train a model, or the model reproduced the protected data in its outputs. Plaintiffs can use Membership Inference Attacks (MIAs) or Dataset Inference (DI) to determine if their data was used for **training**. MIAs (Song & Namiot, 2023) identify if *a single* data point was used, while DI (Maini et al., 2021) shows if *any* data from a set was used, though it doesn't specify how many or which; the choice between them depends on the requirements of a particular lawsuit. Both methods rely on differences between models trained with and without the data. However, MIAs typically require at least access to the model architecture to train reference models, which is often unavailable for proprietary models. Furthermore, existing high-performing MIAs rely on training *several* reference models, a process infeasible with SotA models due to their size. Proving infringement through **data reproduction** is equally challenging. Generative models, by their probabilistic nature, have a non-zero chance of producing any output, raising the question of how likely "too likely" is to constitute infringement. A potential baseline is to compare the probability of the contested output in the suspect model against a model known not to have been trained on the data. Even more problematic are near-duplicates: determining what constitutes a near-duplicate and how to account for *all possible* variations remains a serious problem.

### 5.1. DP for Copyright Protection

DP addresses the challenge of near-duplicates by ensuring that the probabilities of generating copyrighted content *and its variations* remain nearly unchanged through the inclusion of any copyrighted work. Privacy auditing should then be employed to verify adherence to agreed privacy levels. Established minimal privacy levels could be formalized through legislation, allowing copyright holders to negotiate

---

[1]A different set of cases examines whether AI generated outputs can be protected by copyright. We do *not* discuss these cases here.

for more lenient levels in exchange for compensation.

## 5.2. Prior Work

Near Access-Freeness (NAF) bounds a model's output to avoid unauthorized reproduction of copyrighted material (Vyas et al., 2023). Informally, NAF keeps a model's output distribution within a small divergence of a "safe" generative model trained without access to the copyrighted elements. While NAF shares conceptual similarities with DP, there are differences: DP ensures that a model reveals (almost) nothing about the presence of individual samples, which is a *stronger guarantee* than NAF's, which only requires that the model does not produce outputs excessively similar to copyrighted works. Through this, DP ensures that both possible kinds of copyright infringement are avoided.

## 5.3. Open Questions and Challenges

- **MIA Feasibility:** Are MIAs and DI reliable and efficient enough to attack even the largest SotA models? Are they effective for *all* samples?
- **NAF Auditing:** Can we construct attacks similar to MIAs to determine the NAF levels in place?

# 6. Data Bottleneck

The total amount of publicly available data on the internet is projected to soon be fully utilized for model training. Villalobos et al. (2024) estimate that the size of datasets to train future LLMs will match the volume of text data available online by the end of this decade. This is concerning because data has been identified as a key driver behind model performance (Hestness et al., 2017; Kaplan et al., 2020). To acquire additional data, we may consider synthetic data; however, its usefulness is heavily debated. On the one hand, synthetic data has proven successful in enhancing the performance of models in both text and vision tasks (He et al., 2022; Lu et al., 2023). On the other hand, Geng et al. (2024) suggest that training on the original data instead of leveraging it to train a synthetic data generator produces better downstream results. Furthermore, using data generated by GenAI to train the next generation of models initiates a degenerative process that results in a loss of information about the original data distribution and, ultimately, a deterioration in the quality of downstream models trained on that data (Shumailov et al., 2024). This process is accelerated by the growing adoption of GenAI technology and the resulting pervasion of internet data with GenAI-produced content. Due to the uncertain usefulness of synthetic data, we focus on tapping into *non-synthetic*, *human-generated* data, specifically private data and datasets curated by professional providers. We examine the barriers that discourage individuals from sharing their private data and explore why the emergence of professional data curators remains limited.

**Individuals** are often reluctant to share their information due to privacy concerns, as personal data is sensitive, and past privacy breaches have made people wary about entrusting companies with it (Anant et al., 2020). The limited number of **data providers** stems from fundamental challenges inherent to selling data products: Data is non-excludable; once sold, it can be easily shared or resold without the seller's control, eroding their ability to profit. This creates a reliance on trust between buyers and sellers: Buyers must trust that sellers will not misuse or redistribute their data, while sellers must ensure that the data provided delivers tangible value once deployed. Additionally, the intrinsic difficulty in assessing the quality and utility of data prior to purchase complicates the pricing process (Fricker & Maksimov, 2017; Cosgrove & Kuo, 2020).

## 6.1. DP as a Pricing Scheme

We envision a pricing scheme based on DP, where *stricter privacy protections (and lower utility) result in lower prices, while weaker protections (and higher utility) lead to higher prices*, thus compensating users according to their loss of privacy. This approach not only introduces a novel dimension to data trading but also inherently mitigates privacy concerns. By leveraging DP in pricing, the scheme creates a mutually beneficial framework for all stakeholders involved. **Sellers** can adopt a versioning strategy (Shapiro & Varian, 1998) that allows them to offer data at varying privacy levels, optimizing revenue through market segmentation. Furthermore, even if buyers copy and share purchased data, versions with weaker privacy guarantees remain valuable. Additionally, sellers can enter the market with highly private data and gradually introduce less private versions, allowing them to gauge market response and buyer trustworthiness. **Buyers** are provided with an opportunity to evaluate data quality through cheaper, more private versions before investing in more expensive, higher-utility options, helping to address challenges in data valuation. New purchasing strategies can benefit buyers, such as acquiring a wider variety of highly private datasets instead of fewer, less private ones.

## 6.2. Prior Work

Li et al. (2014) explore a concrete pricing scheme with linear queries using DP where query prices depend on the desired precision (noise level). Their scheme involves data owners supplying their data to market makers, who sell linear queries to buyers. Data owners are compensated based on their data's contribution to a query and the precision. Niu et al. (2018) extend the scheme to dependent data—where data points are not independent and may influence or reveal information about one another due to correlations or relationships. Notably, data owners receive compensation for privacy loss through dependencies with other samples, even if their data was not used in a query.

### 6.3. Open Questions and Challenges

The most critical question is where and how to apply DP, as this decision shapes more than just the privacy-utility trade-off. Options are:

- **Local Differential Privacy:** LDP (Section 2) foregoes the need to trust a central aggregator. However, can high-performing models be trained on locally-private data?
- **In-the-clear Data Sharing with Privacy Auditing:** This option involves sharing unprotected data with a contractual agreement that buyers adhere to privacy-preserving practices, verified through privacy auditing (Steinke et al., 2023). Can we ensure compliance even in *adversarial* scenarios where buyers attempt to cheat? Moreover, is privacy auditing practical and scalable for SotA models?
- **Centralized DP with Trusted Data Holders:** Here, the data holder manages training internally, avoiding raw data sharing but incurring high computational and operational costs. Key questions arise: Can data holders handle these resource demands, or is a trusted third party, like a market maker, needed? Would companies even agree to share their training routines?

## 7. Hallucinations

GenAI has made remarkable strides in producing written language and images that are, *at first glance*, indistinguishable from real data. Yet, upon closer examination, flaws and inconsistencies in the generated output become apparent. Despite its impressive linguistic and visual capabilities, GenAI often produces factually incorrect content, deviates from user instructions, or includes unsolicited information. We adopt Ji et al. (2022)'s definition of *hallucinations* as "generated content that is nonsensical or unfaithful to the provided source content". Our definition also includes content inconsistent with established facts, sometimes considered a separate issue called "factuality" (Wang et al., 2023a). Hallucinations can severely impact both end users and developers by undermining trust, harming reputations, and posing risks, such as generating incorrect medical advice. The eloquence of GenAI's textual outputs can exacerbate this issue, as humans tend to associate eloquent language with credibility (Rogers & Norton, 2010).

**Causes of Hallucinations.** Numerous factors contribute to hallucinations in GenAI models (Ji et al., 2022). In the following, we focus on *data-related* causes due to their direct relevance to DP. Specifically, we examine three examples that effectively demonstrate the applicability of DP in addressing hallucinations.

**Immemorization** is a phenomenon where, despite seeing certain information during training, a model has not stored it in its parameters. While the introduction of DP aggravates this problem by restricting the amount of information that

can maximally be extracted from a single data point, relying *solely on the input context* rather than parametric knowledge is *desirable* for some NL applications. For such tasks, the inability to memorize specific information from training data turns into a strength. **Source-reference divergence** is a fundamental issue that arises when there are discrepancies between input sources and target references in training data. For example, if an image has a mismatched caption that includes details not in the image, training an image captioning model on such data may teach it to invent unnecessary or false information. Assuming that the number of faulty samples is or can be limited, the use of DP guarantees that a model that is trained on the erroneous samples is still (almost) the same as one that was trained only on clean data. **Shortcut learning** can be induced by duplicate samples in the training corpus. It teaches a model to memorize specific phrases instead of considering the context. As the post-processing property of DP extends its guarantee to the probability of generating entire sequences, DP potentially discourages shortcut learning of this kind altogether.

### 7.1. DP for Improved Modularity

To reduce the prevalence of hallucinations, recent advances in AI systems prioritize modularity—designing systems as independent, interchangeable components—over monolithic architectures. For example, Retrieval-Augmented Generation (RAG) separates knowledge retrieval from language generation, allowing models to query external databases instead of storing all information internally (Gao et al., 2023). This approach is similarly reflected in agentic systems, where specialized agents collaborate to tackle complex tasks (Wang et al., 2024a). We conjecture that DP encourages modularity in ML models, enabling them to learn general patterns while remaining "fact-free". In language, for example, *linguistic* patterns are pervasive in the dataset, while *factual* knowledge is limited to a few samples. Careful tuning of privacy parameters may allow models to acquire language structure without memorizing specific facts, aligning with the goals of RAG.

### 7.2. Prior Work

To the best of our knowledge, there is no prior work on the interplay between DP and the mitigation of hallucinations.

### 7.3. Open Questions and Challenges

- **Immemorization:** How much does DP increase immemorization, and how does this impact model performance?
- **Fact-free models:** Can the combination of an external knowledge database and a "fact-free" model mitigate hallucinations even more than normal RAG?

## 8. Interpretability

We adopt Rudin et al. (2021) definition of interpretable ML models as "obeying a domain-specific set of constraints that allow it to be more easily understood by humans." Interpretability allows users to understand how a model reaches its conclusions and what factors influence its decisions, fostering trust and enabling informed decisions about its reliability (Molnar, 2020). It is critical for ensuring fairness, as it helps verify whether sensitive attributes like race, age, or gender were inappropriately used in decision-making (Marcinkevics & Vogt, 2020). Additionally, interpretability aids in debugging by revealing why a model fails, enabling targeted improvements, and creating a feedback loop for refining both the model and data. It also supports knowledge transfer by leveraging insights from past challenges.[2]

### 8.1. DP as an Interpretability Constraint

DP can be viewed as an interpretability constraint because it limits the solutions to a ML problem to those that comply with specific privacy guarantees. For instance, under strict privacy guarantees, private models can only learn patterns that are *common across all samples* in the dataset. As these guarantees are relaxed, models can capture patterns *within subgroups* of the data. We argue that more private solutions are inherently simpler and thus easier to interpret, facilitating human understanding. To demonstrate how this approach aids in interpreting both private and non-private models, we sketch out the following workflow: First, we train a model with strong privacy guarantees and thoroughly analyze its behavior—a process that is simplified when combining DP with inherently interpretable models. Next, we incrementally relax the privacy constraints and train a new model. By contrasting this model to the previous one, we can isolate and focus on the newly learned patterns, avoiding distractions from what was already understood. This stepwise approach allows us to systematically strip away the "knowns" and concentrate on the differences, making it easier to interpret the model's evolution. Through the iterative relaxation of privacy guarantees, we eventually build a comprehensive understanding of a fully non-private model.

### 8.2. Prior Work

Overall, limited work exists analyzing the ramifications of DP training on models. Prior work's analysis of private models is superficial, performing limited visual analyses of learned visual concepts. Harder et al. (2020) train a piecewise linear model with DP-SGD and analyze the learned filters, remarking that the interpretability of their filters diminishes with increased privacy. Another line of work fo-

---

[2]We prioritize interpretability over explainability in our work. Still, we encourage further exploration of the connection between DP and explainability.

cuses on privacy-preserving model explanations and attacks facilitated by nonprivate explanations (Nguyen et al., 2024).

### 8.3. Open Questions and Challenges

- **Impact of DP on Learning Behavior:** The application of DP has implications for the general learning behavior of ML models. What are the consequences of employing DP as an interpretability constraint? To explore this, researchers can address several sub-questions:
  - Are certain features or circuits less frequent or absent under DP? This can be analyzed using autoencoders (Bricken et al., 2023) or probes (Zhao et al., 2024).
  - Does DP training reduce or amplify behaviors like sycophancy or biases?
  - Does DP influence phenomena like grokking (Power et al., 2022), particularly the transition from memorization to generalization?
- **Behavioral Consistency across Privacy Levels:** Do insights from models with strong privacy guarantees apply to those with weaker guarantees, or does behavior significantly change when privacy protection is reduced?

## 9. Alternative Views

In this section, we present an opposing position to our own that highlights the practical shortcomings and conceptual limitations of DP when applied to mitigate GenAI issues.

### 9.1. Practical Limitations

**Performance Gap.** The competitiveness of private LLMs remains uncertain (Tramèr et al., 2022). Despite promising advances in private image generation (Ghalebikesabi et al., 2023), performance still trails nonprivate models, though the gap is narrowing in other domains. The adoption of the pretraining-on-public-data paradigm (Abadi et al., 2016), along with the realization that private training requires different hyperparameters (De et al., 2022) and the recent development of new ML training algorithms (Kairouz et al., 2021) have led to significant improvements in the performance of private models. Theoretical advances in DP variations have improved privacy loss estimation over training epochs, enabling equal privacy guarantees with less noise.

**Training Efficiency.** Training with DP-SGD faces *extended training times* and *high memory demands* due to DP-SGD's requirement to compute per-sample gradients. The issue is even more pronounced when training SotA GenAI systems, where both training durations and memory requirements are already considerable for nonprivate models. However, recent advancements in ML frameworks, such as just-in-time compilation and vectorization, have greatly accelerated training processes (Subramani et al., 2020). Similarly, innovations like ghost-clipping (Li et al., 2021) trade off slight

computational complexity gains for reduced memory usage.

**Parameter Selection.** DP establishes an upper limit on the success rate of MIAs, enabling its parameters to be set based on the desired level of protection. However, while revealing an individual's presence in a cancer prediction dataset is sensitive, revealing membership in a dataset containing nearly the entire Internet is not. In these situations, guarantees against more relevant types of attacks take priority. ReRo (Balle et al., 2022) provides a limit on how accurately training data can be reconstructed, based on a specific loss function. This helps to make DP's guarantees more tangible, addressing a common gap in understanding (Cummings et al., 2021). However, measuring recognizability using loss functions remains challenging, as they often fail to align well with human perception (Alva-Manchego et al., 2021; Sun et al., 2024).

### 9.2. Conceptual Limitations

**Inappropriate Privacy Units.** For Internet data, it may be difficult to accurately map data samples to individuals; however, even that may still fall short of protecting privacy. Linguistic exchanges and images may expose information about third parties who are neither part of the conversation nor present in the content. Additionally, privacy protections diminish as more users reference the same information, meaning frequent mentions reduce protection, contrary to common privacy expectations. Consequently, user-level privacy provides inadequate privacy protection. Instead, it is more appropriate to focus on the protection of secrets (Brown et al., 2022). However, implementing such protections demands a sophisticated understanding of natural language and the nuanced dynamics of human interactions– an ambitious and technically demanding prerequisite.

**Comparison with Contextual Integrity.** Contextual Integrity (CI) (Nissenbaum, 2004) is a privacy framework that conceptualizes privacy as the "appropriate" flow of information, where the appropriateness is determined by contextual norms. These norms define information flows in terms of the sender, recipient, subject, information type, and transmission principles. In contrast to CI, DP is a context-agnostic framework, treating all data uniformly. While this simplifies implementation and analysis, it also highlights key limitations. DP is adept at enforcing *negative* privacy rules, effectively preventing the exposure of sensitive information, but lacks the nuance to support *positive* rules—enabling the flow of information when contextually appropriate. This rigidity can lead to overly restrictive privacy mechanisms that fail to accommodate real-world complexities. Conversely, CI excels in its nuanced understanding of context-specific norms, aligning more closely with human expectations around privacy. Yet, CI offers no clear operational mechanisms to enforce or measure appropriate information flows (Ben-

thall, 2021). Recent efforts synergize the two frameworks, promising guidance for DP parameters and adding another dimension to CI in the form of a "transmission property" (Benthall & Cummings, 2024).

## 10. The Path Forward

Throughout this paper, we have explored the potential of DP to address key challenges in GenAI. Simultaneously, we have identified both practical and conceptual limitations that demand further investigation. Where does this leave us?

**Untapped Opportunities**. We have demonstrated that DP offers untapped opportunities that are ripe for exploration. Our aim has been to inspire readers to address the open questions we have highlighted throughout this work. From refining current approaches to tackling entirely new challenges, the research landscape remains vast and promising.

**Defining Meaningful Privacy**. A critical next step is to determine the levels of DP that are necessary to guard against *meaningful* attacks. The emergence of Reconstruction Robustness (ReRo) has taken strides in this direction by shifting focus from basic membership inference to data reconstruction. The task now is to understand what these bounds can protect against: What level of privacy is required to render images unrecognizable? What meaningful information can we still garner from reconstructed texts? Some works (Ziller et al., 2024; Schwethelm et al., 2024) have already taken up the challenge.

**Adopting DP's paradigm**. DP's worst-case guarantees may exceed the privacy requirements of specific use cases. Since DP's protections come at the price of reduced utility, it may be necessary to adopt tailored guarantees that strike a better balance for the task at hand. Even when diverging from DP's exact definition, starting from its framework and stripping away unnecessary protections can inspire effective, context-specific solutions.

**Resolving Conceptual Challenges**. We must confront the conceptual challenges that DP faces, particularly in unstructured domains like natural language and images. Originally designed for structured data, DP's application to these areas remains fraught with unresolved questions. How can we adapt DP to handle the complexities of language and visual data effectively? Additionally, the integration of DP with frameworks like CI warrants deeper investigation.

## Impact Statement

This paper addresses critical issues in GenAI, offering insights and potential solutions that can contribute to the development of more robust, ethical, and reliable generative models. By tackling challenges such as hallucinations, privacy concerns, and copyright infringement, this research

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

# A. Backdoor Attacks

Backdoor attacks represent a sophisticated form of attack on GenAI systems.[3] They introduce latent capabilities and behaviors, triggered only under specific conditions, into the model by inserting manipulated samples into the training data (Goldblum et al., 2020). The goals that can be achieved using backdoors can be incredibly varied and complex: In text generation, a backdoor might cause an AI to suggest vulnerable code snippets at a higher-than-average rate (Hubinger et al., 2024) or to express biased opinions on specific topics (Li et al., 2024). In image generation, backdoors can systematically embed subtly biased visual representations, influencing users' perceptions (Vice et al., 2024).

Backdoors are highly dangerous due to their subtlety; they can activate in specific contexts, like outside testing environments or at certain times, often evading detection through traditional security assessments. Attempts to remove backdoors may inadvertently refine a model's ability to detect triggers, exacerbating the risk (Hubinger et al., 2024).

Backdoor attacks pose a significant threat to GenAI systems, which typically rely on vast amounts of internet-sourced data. The openness of online content makes it relatively easy for malicious actors to introduce harmful inputs (Carlini et al., 2024). Compounding this risk, backdoors are often model-agnostic (Zou et al., 2023), meaning they can exploit vulnerabilities across diverse AI architectures, further exacerbating security concerns.

## A.1. DP as a Defense

Backdoor attacks involve introducing carefully crafted poisoned samples into the training data to induce specific behaviors in a model. Because DP limits the influence of any single sample, attackers may have to introduce a considerably larger number of poisoned samples to achieve the desired effect. This not only raises the cost and complexity of an attack but also makes it substantially easier to detect anomalies in the training data.

## A.2. Prior Work

Most prior work investigating DP as a defense mechanism has focused more broadly on *general* data poisoning attacks. None of these works specifically addresses GenAI.

Ma et al. (2019) demonstrate DP's inherent protection against data poisoning, deriving bounds on the required number of samples necessary to achieve an attacker's goal. (Jagielski et al., 2020; Geiping et al., 2020; Xu et al., 2021) extend the analysis to ML training using DP-SGD for Natural Language Processing (NLP) and computer vision tasks, respectively, confirming DP's effectiveness as a defense.

## A.3. Open Questions and Challenges

- **Impact on Backdoor Reliability**: How does DP influence the reliability of triggering backdoors? What DP levels effectively guard against them?
- **Backdoor Goals and DP**: Given the varied objectives of backdoors, are certain goals easier to achieve than others, and does DP affect this?
- **Detection**: Does DP make backdoor detection more difficult? Since DP bounds both the probability of successful triggering and detection, do these probabilities increase at the same rate, or does one grow faster, favoring attackers or defenders?

# B. Acronyms

**AI**      Artificial Intelligence

**CI**      Contextual Integrity

**DI**      Dataset Inference

**DP**      Differential Privacy

**DP-SGD**  Differentially Private Stochastic Gradient Descent

**GenAI**    Generative Artificial Intelligence

**GDPR**    General Data Protection Regulation

---

[3] In this paper, we focus on backdoors introduced through *data*.

**LLM**      Large Language Model

**LDP**      Local Differential Privacy

**MIA**      Membership Inference Attack

**ML**       Machine Learning

**NLP**      Natural Language Processing

**NAF**      Near Access-Freeness

**PII**      Personally Identifiable Information

**ReRo**     Reconstruction Robustness

**RAG**      Retrieval-Augmented Generation

**SotA**     State-of-the-Art

