# OpenReview forum: "Position: Generative AI and Differential Privacy --- A Perfect Match"
_ICML.cc/2025/Position_Paper_Track — Submitted to ICML 2025 Position Paper Track_

### Official Review · Reviewer_DP9D · 2025-02-19

**Significance:** 4
**Argument Clarity:** 3
**Rating:** 4
**Confidence:** 4

**Questions:**

- Q1 How are you going to revise the paper based on the weaknesses pointed out regarding privacy units and privacy budget (W1) and formal vs. emperical bounds (W2)?
- Q2 Which MIAs are you referring to and does my suggestion make sense in this case? (W3)
- Q3 How do you see the argument of data processing inequality? (W4)
- Q4 Do you think the ethical argument regarding obtaining for money for less privacy makes sense? How are you going to edit the text to refer to it? (W5)
- Q5 How are you going to make the data sharing model clearer in terms of where the DP could be applied? (W6)
- Q6 Is the clash between DP and interprebility and fairness relevant for your work? What are reasons why work is not mentioned? (W7)
- Q7 Are the minor suggestions helpful and are you going to adress all of them? If not which ones are not going to edit in? (minor issues)

**Discussion Potential:**

2

**Paper Summary:**

The position paper argues that the application of Differential Privacy (DP) to Generative Artifial Intelligence (GenAI) can help mitigate more issues than just privacy related issues but the migitation effect regarding these issues when applying DP is underexplored in research. The issue explored in the paper are the memorization of PII, data quality, harmful knowledge, easing challenges related to the right to erasure, Copyright Infringement, data bottlenecks, hallucinations and interpretability. For all issues the potential benefit of DP is outlined, the current state of research is presented and open questions and challenges are discussed. Finally, the paper concludes with presenting alternative views and sketches a potential way forward.

## update after rebuttal
I believe the authors have addressed most questions. I agree with the reviewer `3oGB` that the paper is "thought provoking" and I appreciated the points the authors brought up regarding this. I change my score from 3 to 4 and trust the authors that they will make the appropriate changes as promised in their rebuttal.

> We want to stress that while revisions to the current paper during the review process are not possible, revisions in the published version are possible, and we are glad to incorporate the reviewers’ feedback into the final manuscript.

**Position:**

Yes

**Position In Title:**

Yes

**Related Work:**

2

**Strengths And Weaknesses:**

While the paper is well written, relevant and novel there are quite some details that I think are not correct. With these I don't mean the position itself but important considerations when discussing the position or missed related work. I feel the authors can fix these but the general issue is that I don't think revisions are allowed and the text in an unrevised version is not good enough to be published in ICML.

# Strenghts
- The paper discusses a timely and relevant manner as GenAI is both researched and applied but suffers from critical issues.
- Generally, the suggestion to borrow techniques from DP is well supported by arguements and extensive literature references but sometimes these are outdated or not complete (see weaknesses).
- The motivation and relevance is motivated both by observations from research community as well as from outside references (e.g., court cases and industry reports).
- The paper makes clear what is known (prior research) and what is not (open question/challenges)
- The paper proposes a clear idea on what to do next which makes potential discussions more interesting for the community.


# Weaknesses
- W1 **Too brief background discussion on privacy budget and privacy units**: The paper lacks background discussion on two core DP research that is are essential to mention to make the discussion fruitful for the community (especially as this is targeted at wider audience than DP researchers). These are the: 1. privacy budget (and that just doing DP with, e.g., large $(\epsilon, \delta)$ values cannot considered to be private), see discussion on "performative privacy" in Sarathy (2022) [1] and 2. privacy units where the paper sketches briefly what they are but explains not throughoutly what it means in practice. There is work, e.g., in LLM community [2] and document intelligence community [3] where privacy units that are more complex but also more relastic than item-level neighborhoods are discussed in detail. This aspect seems relevant to discussed more as you specifically discuss this as a key challenge but the implications and complexity of this are not clear from the current manuscript (e.g., in Section 1 you speak of an individuals data point whereas in Section 2 you mention DP-SGD with a single training sample whereas practical deployments would bound the influence of the privacy unit [3])
- W2 Section 3.4: The discussion of formal bounds provided by DP (e.g., the TPR/FPR bounds by Kairouz et al. [4]) and the emperical bounds obtained through privacy attacks (e.g., MIAs in different threat models e.g., Nasr et al. [5]) is not precise enough. DP does not predict bounds either.
- W3 Section 5: The background on MIAs would benefit from an updated discussion as SoTA methods like LiRA [6] and RMIA [7] are somewhat robust to changing the shadow model architecture (Figure 11a in [6] and Appendix C.3 in [7]). RMIA performs also well with fewer shadow models (Figure 3 of [7]). What might be an issue is the general MIA game (see e.g, [6]) that assumes that much of the dataset is known. There might methods that you speak of that don't have this problem but given that there are not references it is hard to judge if you aware of this.
- W4 Section 6: Your discussion of synthetic data does not mention the Data processing inequality at all. It might be worth considering.
- W5 Section 6.1: The obvious counterargument that this leads to users that are in financial problems selling their data for monetary reasons is not discussed and so are potential ethical considerations related to this.
- W6 Section 6.1: It could be written earlier how the data sharing model would work. In Section 6.3 you write that it is unclear where DP should be applied but in 6.1 you are not too clear about this. You could make Section 6.1 more clear by introducing that more carefully.
- W7 Section 8: I believe your discussion misses the observation of prior work that DP makes interpretability (and fairness) harder (see e.g, Section 5 of Ferry et al. (2023) but there are more references). Also Section 8.3 asks if DP increases biases but I believe that has been at least shown with DP-SGD before (e.g., Esipova et al (2023) [9])

Minor suggestions in writing:
- Section 3.4: Performance vs. Harm Mitigation while at least to a DP researcher "utility" is usually the more familar term. Performance is sometimes discussed in terms of computationally efficiency of DP-SGD. Switching to utility or explaining what you mean clearer might help understanding. BTW you use utility in Section 6.3
- Section 5: You cite Song & Namiot for MIAs when the reference is a survey about model inversion attacks. It would be good to check if this reference is appropriate here.
- Section 5.1: Privacy auditing is never introduced nor cited here. It would be good to explain in one sentence what it is and cite an appropriate reference here.
- Section 6.1: Parameter selection might be misunderstood, perhaps adding DP parameter selection or privacy budget might be easier to comprehend?
- Section 7: Missing a lot of references to, e.g. Immemorization, Source-reference divergence, Shortcut learning.
- Section 9: The discussion on training efficiency could benefit from a recent comprehensive study comparing SoTA approaches and overcoming issues with just-in-time-compilation by Beltran et al. (2024) [10]

[1] Sarathy, J. (2022). From algorithmic to institutional logics: the politics of differential privacy. Available at SSRN 4079222.

[2] Chua, L., Ghazi, B., Huang, Y., Kamath, P., Kumar, R., Liu, D., … Zhang, C. (2024). Mind the Privacy Unit! User-Level Differential Privacy for Language Model Fine-Tuning. First Conference on Language Modeling. Retrieved from https://openreview.net/forum?id=Jd0bCD12DS

[3] Tito, R., Nguyen, K., Tobaben, M., Kerkouche, R., Souibgui, M. A., Jung, K., ... & Karatzas, D. Privacy-aware document visual question answering. In International Conference on Document Analysis and Recognition 2024.

[4] Kairouz, P., Oh, S., & Viswanath, P.. The composition theorem for differential privacy. In ICML 2015.

[5] Nasr, M., Songi, S., Thakurta, A., Papernot, N., & Carlin, N. Adversary instantiation: Lower bounds for differentially private machine learning. In 2021 IEEE Symposium on security and privacy (SP).

[6] Carlini, N., Chien, S., Nasr, M., Song, S., Terzis, A., & Tramer, F. Membership inference attacks from first principles. In 2022 IEEE Symposium on Security and Privacy (SP).

[7] Zarifzadeh, S., Liu, P., & Shokri, R. Low-Cost High-Power Membership Inference Attacks. In ICML 2024.

[8] Ferry, J., Aïvodji, U., Gambs, S., Huguet, M. J., & Siala, M. (2023). SoK: Taming the Triangle--On the Interplays between Fairness, Interpretability and Privacy in Machine Learning. arXiv preprint arXiv:2312.16191.

[9] Esipova, M. S., Ghomi, A. A., Luo, Y., & Cresswell, J. C. Disparate Impact in Differential Privacy from Gradient Misalignment. In ICLR 2023.

[10] Beltran, S. R., Tobaben, M., Jälkö, J., Loppi, N., & Honkela, A. (2024). Towards efficient and scalable training of differentially private deep learning. arXiv preprint arXiv:2406.17298.

**Support:**

4

---

> ### Author Rebuttal · Authors · 2025-03-31
>
> We thank the reviewer for their very thorough and in-depth review. We want to stress that while revisions to the current paper during the review process are not possible, *revisions in the published version are possible*, and we are glad to incorporate the reviewers’ feedback into the final manuscript. We address each of your raised points below:
>
> **Lax privacy guarantees offer no practical protection:** While there is evidence supporting your point, the adequacy of privacy depends greatly on what needs to be protected. Certain privacy levels may provide insufficient protection against membership inference attacks; however, they can still offer meaningful safeguards against data reconstruction attacks, as demonstrated by [1]. We will clarify this distinction in the revised manuscript.
>
> **Missing explanation on privacy units:** We acknowledge this omission. Our choice not to elaborate on data units in differential privacy stems from our assessment that these distinctions would not significantly impact our application of DP for the considered use cases.
> In our discussion of challenges for DP in natural language contexts, we highlight that user-level privacy may be unattainable due to the inherent difficulty in linking text to its author. We will include a more explicit explanation of this point.
>
> **Discussion of theoretical and empirical DP bounds not precise enough:** We appreciate this feedback and would welcome more specific advice on enhancing our discussion's precision. As noted in our response to reviewer nmwS, we plan to include a more comprehensive introduction to DP in the appendix, where we would be pleased to incorporate relevant theoretical bounds.
>
> **DP not providing bounds:** We would appreciate further clarification regarding this concern, as differential privacy is fundamentally a framework that provides upper and lower bounds [2] on privacy leakage.
>
> **Updated references on MIAs:** We thank the reviewer for highlighting these additional papers, which we will incorporate into our text; space constraints drove us to cite a survey paper on membership inference attacks rather than these individual works. However, we note that even in the RMIA paper's analysis (Appendix C.3), the methodology relies on utilizing identical architectures for both target and shadow models.
>
> **Data processing inequality, ethical concerns with data trading, and data sharing model clarification:** We will incorporate a discussion of the data processing inequality in the limitations section, address the ethical considerations of data trading in our impact statement, and restructure the explanation of the data sharing model to enhance clarity and comprehensibility. Thank you for helping us catch these important points.
>
> **DP makes interpretability harder:** We appreciate the reviewer's comment regarding the relationship between DP and interpretability. It is important to note that the cited paper appears to conflate interpretability and explainability, a distinction we carefully maintain. Our analysis specifically addresses interpretability. We would like to emphasize that the cited work does not explicitly assert that DP diminishes interpretability. Rather, it suggests that inherently interpretable models may, in non-DP training contexts, be susceptible to leaking training data information.
>
> **DP and biases:** We acknowledge the need for greater specificity regarding our discussion of biases. Our analysis focuses on social biases rather than disparate performance across groups. We will revise the text to make this distinction explicit.
>
> **Minor corrections:** We agree with all suggested minor corrections and will implement them in the camera-ready version.
>
> ### References
> [1] Ziller, Alexander, et al. "Reconciling privacy and accuracy in AI for medical imaging." Nature Machine Intelligence 6.7 (2024): 764-774.
>
> [2] Balle, Borja, et al. Privacy Amplification by Subsampling: Tight Analyses via Couplings and Divergences. arXiv:1807.01647, arXiv, 23 Nov. 2018. arXiv.org, https://doi.org/10.48550/arXiv.1807.01647.

---

> > ### Comment · Reviewer_DP9D · 2025-04-02
> >
> > Thanks to the authors for replying to the questions!
> >
> > **DP not providing bounds**
> > > We would appreciate further clarification regarding this concern, as differential privacy is fundamentally a framework that provides upper and lower bounds [2] on privacy leakage.
> >
> > You write in your text the following:
> >
> > > However, the guarantees provided by DP are often overly conservative, as real-world scenarios tend to yield tighter bounds than
> > predicted by DP.
> >
> > DP does not predict. I have a problem with the word predict here. DP provides theoretical bounds that can be complemented with empirical bounds obtained through attacks (see references posted). You can argue that the theoretical bounds are not tight enough in realistic settings which is true, but in the current form the manuscript is inprecise here.
> >
> > **Updated references on MIAs**:
> >
> > > However, we note that even in the RMIA paper's analysis (Appendix C.3), the methodology relies on utilizing identical architectures for both target and shadow models.
> >
> > This is not true and I encourage the authors to read the complete C.3 Section as RMIA authors write (the following is a direct quote): *"Figure 13 presents the performance of attacks when different architectures are used to train reference models, while keeping the structure of the target model fixed as WRN28-2. So, the target and reference models may have different architectures."*
> >
> > **DP makes interpretability harder:**
> > >  We appreciate the reviewer's comment regarding the relationship between DP and interpretability. It is important to note that the cited paper appears to conflate interpretability and explainability, a distinction we carefully maintain.
> >
> > I apparently did not understand your manuscript then fully. Could you please provide me with your definitions of both so that I can follow. BTW this might be then also quite important for the manuscript.

---

> > > ### Author Response · Authors · 2025-04-06
> > >
> > > Thank you for your swift reply and continued engagement with our work. We will address your points following the order in your comment.
> > >
> > > **Incorrect use of "predict":** We acknowledge your concern about our phrasing. You are correct that differential privacy provides theoretical bounds rather than predictions. We will change our wording to reflect your suggestion.
> > >
> > > **Membership Inference Attacks:** You are right; this was an error on our part. We apologize for the mistake and will update our statement regarding the architectural requirements for MIAs.
> > > However, it's worth noting that the RMIA authors themselves state that "the optimal performance for attacks is observed when both target and reference models share similar architectures."
> > >
> > > Most importantly, our central argument remains: even under identical model architectures, MIAs do not reliably detect the presence or absence of samples in training datasets with sufficient accuracy to raise practical privacy concerns, except in auditing settings. We will revise our manuscript to emphasize this critical point more clearly.
> > >
> > > **Interpretability and explainability definition confusion:** In our paper, we adopt the definition of interpretable machine learning proposed by Rudin et al. [1], which describes it as "obeying a domain-specific set of constraints that allow it to be more easily understood by humans." We consider differential privacy as one such constraint that makes interpretation easier for humans. Although we do not explicitly state it in the text, our definition of explainability includes post-hoc methods that provide justifications for model outputs without necessarily disclosing the internal mechanics of the model. While these definitions are well-established in the field ([2], [3], [4], [5]), we recognize the importance of emphasizing the distinction between the two. We will update our manuscript to highlight this differentiation.
> > >
> > > ## References
> > > [1] Rudin, C., Chen, C., Chen, Z., Huang, H., Semenova, L., & Zhong, C. (2022). Interpretable machine learning: Fundamental principles and 10 grand challenges. Statistic Surveys, 16, 1-85.
> > >
> > > [2] Rudin, Cynthia. “Stop Explaining Black Box Machine Learning Models for High Stakes Decisions and Use Interpretable Models Instead.” Nature Machine Intelligence, vol. 1, no. 5, May 2019, pp. 206–15. DOI.org (Crossref), https://doi.org/10.1038/s42256-019-0048-x.
> > >
> > > [3] Bommasani, Rishi, et al. "On the opportunities and risks of foundation models." arXiv preprint arXiv:2108.07258 (2021).
> > >
> > > [4] Turpin, Miles, et al. "Language models don't always say what they think: Unfaithful explanations in chain-of-thought prompting." Advances in Neural Information Processing Systems 36 (2023): 74952-74965.
> > >
> > > [5] Chaddad, A.; Peng, J.; Xu, J.; Bouridane, A. Survey of Explainable AI Techniques in Healthcare. Sensors 2023, 23, 634. https://doi.org/10.3390/s23020634

---

### Official Review · Reviewer_3oGB · 2025-02-25

**Significance:** 3
**Argument Clarity:** 2
**Rating:** 3
**Confidence:** 3

**Questions:**

Have the authors considered experimenting with DP for any of the less common use cases?

What under-explored use case do the authors think DP would be most beneficial for, and why?

**Discussion Potential:**

3

**Paper Summary:**

The paper proposes that Differential Privacy (DP) could be a potentially viable tool beyond privacy. It might also apply to memorization-based copyright, harmful content generation, data quality harms, the right to erasure, hallucination and interpretability. In essence, the paper argues that DP’s objective, ensuring a single data point has negligible impact on a model output, is clearly applicable to a range of issues, and is under-explored in its potential. They propose directions of future work which would help uncover this potential and its viability in relation to other methods/trade-offs.

**Position:**

Yes

**Position In Title:**

Yes

**Related Work:**

2

**Strengths And Weaknesses:**

This work’s proposition has a few notable strengths:
* The idea is simple and elegant: that DP as a tool might generalize to address many problems in tandem.
* The write-up is clear, and creatively explores potential implications of such methods and their interactions with other methods, eg. machine unlearning, membership inference attacks, etc..
* The authors have thought broadly, aggregating and impressive range of problems that might be impacted by some variants of DP, such as copyright, right to erasure, etc..
* The proposal focuses on preventative rather than reactive methods to these identified problems. It also poses important questions to its own proposition, that examine the implicit cost of such methods to system capabilities performance, or to trade-offs between (or complementarity of) proactive and reactive methods.

There were also some places were the work could have been strengthened:
* The main proposition might have been more thoroughly supported by evidence or empirical work. While there is discussion of prior work after each subsection, it rarely dictates the extent to which DP would actually mitigate that problem. Without this, the paper reads mainly like a series of proposals for future work, with some speculation as to the potential benefits of a method.
* It would greatly benefit the work to at least conduct some initial experiments, for at least one of the identified problems that is not normally addressed by DP. Basic experiments could at least help assess DP’s viability and bring more awareness to the challenges in its implementation for unique problems.
* Even without experiments, the work could explore fewer problems more rigorously, to understand their viability, rather than more problems less rigorously. For instance, while DP has been used effectively for privacy/PII, the case for data quality seems under-defined and ambiguous. To what extent are **infrequent** inaccuracies from the web actually a central driver of misinformation in model outputs? What about other notions of “data quality”? Might this approach also act to remove lesser represented perspectives?
* The impact of DP on hallucinations and interpretability, or its use as a “pricing scheme”, especially, appear very speculative in their methodology and impact of their application.
* The copyright section is mostly well reasoned, but omits a key discussion on the distinction between model inputs and outputs. DP would not seem to solve the copyright infringement allegations connected to copying data (even before training)? They would only have the potential to mitigate model output-based infringement allegations. Even then, copyright infringement allegations are often for data which is frequently repeated on the web. This might severely limit DP, but there is little mention of these limitations in the discussion.

Overall, the authors propose an elegant and interesting idea, that is certainly thought provoking. But at the same time, there is a dearth of rigor and depth in the assessment of the proposal. The work can be read as an attempt to shoe horn the same “hammer” for every nail. It would benefit from deeper discussion of the potential pitfalls or shortcomings of this tool for some of the use cases.

**Support:**

1

---

> ### Author Rebuttal · Authors · 2025-03-31
>
> We thank the reviewer for their thorough assessment and constructive feedback on our manuscript. We address each point raised below.
> ### Weaknesses
> **Lack of evidence and no initial experiments:** We acknowledge that initial experiments would strengthen our position paper. However, our paper is *intentionally* designed to promote further research into using differential privacy for issues related to generative AI. The ICML call for papers explicitly requests submissions that adopt a broader perspective and offer recommendations for future actions.
>
> **Should have investigated fewer issues but more deeply:** Our deliberate choice of breadth over depth stems from the paper's aim to illustrate the versatility of differential privacy across multiple domains. Rather than developing deep but narrow technical contributions, we sought to establish conceptual connections across fields, thereby highlighting potential applications that merit further investigation.
>
> **To what extent are infrequent inaccuracies from the web actually a central driver of misinformation in model outputs?** This question merits dedicated research beyond the scope of our current work. While we cannot quantify the relative contribution of web inaccuracies to model misinformation, evidence indicates it contributes to the broader problem. Our proposal to mitigate this through differential privacy represents one approach to addressing this identified contributor.
>
> **What about other notions of data quality?** Our framework intentionally maintains flexibility regarding definitions of data quality. The sample-based privacy parameter selection methodology we propose can accommodate various quality metrics, allowing researchers to implement the approach using domain-specific quality criteria most relevant to their applications.
>
> **Might this approach also act to remove lesser-represented perspectives?** We concede that our discussion inadequately addresses this important concern and commit to incorporating this caveat into the camera-ready version.
>
> **DP's application in hallucination mitigation, interpretability, and data pricing is too speculative:** While we acknowledge the speculative nature of some applications discussed, we note that this aligns with our paper’s intent (see **Lack of evidence** above). Regarding data pricing specifically, prior work exists investigating this use case.
>
> **DP does not prevent data copying:** We kindly refer the reviewer to our reply to reviewer nmwS.
>
> ### Questions
> **Have the authors considered experimenting with DP for any of the less common use cases?** After completing the position paper, we, and hopefully many readers, will be eager to explore the research directions outlined.
>
> **What under-explored use case do the authors think DP would be most beneficial for, and why?** We believe that each use case we consider in the paper is valuable. Personally, we like to work on both more and less speculative topics at the same time to strike a good balance. We regrettably cannot elaborate further on our topic selection to maintain anonymity.

---

### Official Review · Reviewer_nmwS · 2025-03-13

**Significance:** 3
**Argument Clarity:** 3
**Rating:** 4
**Confidence:** 3

**Questions:**

I apologize in advance to the authors for the question, which my be due to my lack of understanding.

I am referring to the last sentence of section 5.2: "..., DP ensures that both possible kinds of copyright infringement are avoided"

It is not clear to me why DP ensures input and output infringements are avoided.
DP masks a copyrighted input so as not to replicate it as an output. Nevertheless, the copyright violation did occurr at the input step, and while the input item won't show up as an output, yet it was part of the training.

**Discussion Potential:**

4

**Paper Summary:**

The paper makes the case for an integration of the Differential Privacy (DP) paradigm into the structure of GenAI, and how DP can help manage several current issues related to the development, implementation, and use of GenAI.
The view of the authors is that DP is more than a privacy protection tool. Indeed, their claim is "The goal of our paper is to underline DP’s untapped potential and inspire deeper exploration into its applications within GenAI."

They  then proceed to discuss a list of identified issues in GenAI that can be addressed with DP.
- Proactive Measures: memorization of Personal Identifiable Information (PII), data quality, and harmful knowledge
- Right to Erasure
- Copyright Infringement
- Data Bottleneck
- Hallucinations
- Interpretability
- Backdoor attack (Appendix)
For each item they provide a short and clear description of the issue, the current SotA, and the high level description of improvement "expected" by leveraging the DP approach. They conclude each item with a short list of Open Questions and Challenges.

In the final section of the paper, the authors list and address some of the opposing positions. They discuss two main limitations of DP.
- Practical Limitations, such as performance, training efficiency, and parameter selection not aligned with human perception.
- Conceptual Limitations:  indirect contextual information or linguistic exchanges may effectively challenge/decrease privacy protection. Moreover, DP is based on a context-agnostic framework, and it may be overly restrictive if compared to other privacy frameworks based on contextual norms, such as Contextual Integrity.

In the final section, authors push for adopting the DP paradigm as they aknolewdge the challenges ahead for DP framework.


## update after rebuttal
I have read the rebuttal by the authors, and confirm my scores.

**Position:**

Yes

**Position In Title:**

Yes

**Related Work:**

4

**Strengths And Weaknesses:**

Strengths
The position is clearly stated and supported through the paper by a high level reasoning.
The layout is straightforward.
The topic, Privacy, is of great interest to ICML, and the paper is addressing alternative approaches.

Weaknesses
I feel the reader could benefit from more details on the DP, as this paper is intended to address a wide audience.
As a consequence of the previous line, I feel that some of the potential improvements delivered by DP are discussed at too high level, I do understand authors rightly wanted to list as many issues/items as possible to make their case, but that prevented more details which could have made the position stronger.

**Support:**

4

---

> ### Author Rebuttal · Authors · 2025-03-31
>
> We sincerely thank the reviewers for their valuable time and constructive feedback. Please let us address your addressed weaknesses and answer your questions.
> ### Weaknesses
> **The high-level introduction to DP:** We acknowledge that the current introduction to DP is somewhat cursory. We concur that the paper would benefit from a more comprehensive presentation of DP. The brevity in the original submission was primarily motivated by space constraints. To address this, we will incorporate a more detailed exposition of DP into the appendix of the camera-ready version.
> ### Questions
> **How does DP help with (input) copyright infringement?** We thank the reviewer for raising this question, as it highlights an important potential misunderstanding that we aim to clarify both in this rebuttal and in the revised paper. Differential Privacy limits any single data point's influence on the output, making detecting a sample's presence in the training data statistically (nearly) impossible. Therefore, we argue that if we cannot reliably detect its presence, even if a copyrighted sample was included in the training data, we should consider treating the sample as if it were not part of it. The post-processing property of DP then ensures that the privacy guarantees extend to the model outputs.

---

### Decision · Program_Chairs · 2025-04-27

**Decision:**

Reject

**Comment:**

This work claims that DP has an untapped potential to resolve many of the challenging issues in GenAI. It then lists issues and discusses how these issues would be addressed by using DP.
The main issue of this work is basically that it is a mix of issues that DP is designed to address and it has already motivated plenty of ongoing work to train GenAI with DP. These are all things related to data memorization (privacy, copyrights, unlearning, data poisoning, a way to get access to more data).
The second set of issues are hallucinations and interpretability where the arguments are weak and, in my opinion, provide no credible evidence that DP would in any way address the hallucinations or make the models more interpretable.
Overall, while I certainly agree with the main conclusion that DP has many benefits (including those beyond privacy) and that more work needs to be done on training models with DP I think this paper does not add anything new to this discussion.